# Reinforcing Behaviors of Sulfur-Containing Silane Coupling Agent in Natural Rubber-Based Magnetorheological Elastomers with Various Vulcanization Systems

**DOI:** 10.3390/ma13225163

**Published:** 2020-11-16

**Authors:** Mengxin Wang, Xiaoqian Hao, Wenju Wang

**Affiliations:** 1National Key Laboratory of Transient Physics, Nanjing University of Science and Technology, Nanjing 210094, China; menxinwang1982@163.com; 2School of Science, Nanjing University of Science and Technology, Nanjing 210094, China; 220113011215@njust.edu.cn; 3School of Energy and Power Engineering, Nanjing University of Science and Technology, Nanjing 210094, China

**Keywords:** magnetorheological elastomer, silane coupling agent, vulcanization systems, dynamic mechanical behavior, sulfur contribution effect

## Abstract

Magnetorheological elastomers (MRE) is known as an intelligent material constituted of a rubber matrix as well as soft magnetic particles. Silane coupling agents are used to raise the interplay between the inorganic particles and rubber matrix. Silane coupling agent, bis-[-3-(trimethylsilyl propyl)tetra sulfide] (Si69), was picked for comparison of its reinforcing efficiency in the MRE with various vulcanization systems: a conventional (CV), semi-efficient (semi-EV), and efficient (EV) vulcanization system. The outcome illustrated that not only was there improved Si69 surface hydrophobicity of the magnetic particles, but also enhanced Si69 in the interplay between the rubber matrix and magnetic particles. On one hand, the saturated induced magnetic modulus and zero magnetic field modulus of MRE was increased in the vulcanization system, and the loss factor was reduced after the magnetic particles were modified by Si69. On the other hand, the effect of Si69 on the MRE depended on the vulcanization system. The Si69 provided better enhancements in the EV system due to effects of the sulfur contribution of Si69.

## 1. Introduction

Magnetorheological elastomers (MREs) are largely composed of a rubber matrix and soft magnetic particles [1,2]. MREs belong to a new type of intelligent material since the rheological properties can be transformed reversibly and rapidly by applying external magnetic fields [3,4,5]. Typically, components of MREs are cured under a strong magnetic field. The chain-like structure is built by magnetic particles within a rubber matrix after curing, and fixed in the rubber matrix [6,7]. In recent years, MREs have been measured for submissions such as adaptive tuned vibration absorbers [8], engine vibration isolators [9], automotive engine mounts [10], etc. due to their excellent variable stiffness and variable damping characteristics.

It has been illustrated that the property of MRE devices depend on the characteristics of rheological MREs. MREs are considered as particle reinforced composites [11]. Enhancing the interface interaction between the particle and rubber matrix is a way of effectively enhance the mechanical properties of composites. If the particle is a MRE, then it could affect the magnetorheological effect (MR effect). However, the hydrophilic character of the surface of magnetic particles makes them incompatible with the general hydrophobic rubber matrix [12]. The poor interaction leads to disappointing mechanical properties between the particle and matrix. Meanwhile, poor dispersion of particles in the matrix, thereby affecting the characteristics of magneto-rheological damping [13,14]. Therefore, it is reasonable to decorate the surface of magnetic particles and ameliorate the interface within the rubber matrix between interaction and dispersion.

Surface modification of inorganic particles including magnetic particles have been achieved by using methods such as polymer coatings [15], zwitterionic [16] and bifunctional coupling agent treatments [17,18]. In numerous modification approaches, silane-based bifunctional coupling agents have been successful and cost-effective. These agents could promote the interaction between inorganic particles and the rubber matrix. Coupling agents contained a hydrolyzable group of hydrolyze (alkoxy) at one end that would interact with the hydroxyl group on the surface of inorganic particles and organic functional groups at the other end such as an amino group, a vinyl group, and sulfuric atoms, which react with the rubber [19,20,21]. The result showed that the silane coupling agents acted as a bridge within the inorganic particles and rubber matrix, promoting the interaction and dispersion of particles.

Aside from the interfacial interaction between the particles and matrix, it also exhibited an influence of the mechanical properties and MR effect of the MRE. The rubber materials’, final properties were achieved after curing. During the curing process, the molecular chains of rubber reacted with a suitable curing system, which resulted in the formation of cross-linked three-dimensional structures within the rubber matrix [22]. Accelerated systems of sulfur vulcanization are the most ancient and diffusely used curing systems in the rubber industry [23], where they can be categorized as conventional (CV), semi-efficient (semi-EV), and efficient (EV) vulcanization systems [24]. The section of CV, semi-EV, and EV systems depend on the ratio, where the accelerator and sulfur (A/S) is between 0.2–12 [25]. During sulfur vulcanization, different types of sulfide crosslinks between rubber molecular chains are formed [23]. Sulfide crosslinks are connected to the mechanical property of rubber materials such as the dynamic mechanical properties, hardness, and elasticity. Structures might affect the mobility of the magnetic particles of MREs.

Although extensive work has reported the function of silane coupling agents and the influence on the properties of MREs, little attention had been given to the reinforcing efficiency of silane coupling agents with various systems of vulcanization. The effect of a sulfur-containing silane coupling agent and vulcanization system on natural rubber-based MREs filled with carbonyl iron powder were discussed in this work. The bis-[-3-(ethoxy silyl)-propyl]-tetra sulfide (Si69) was used as a silane coupling agent for modified carbonyl iron powder. The effects of silane on the hydrophobic surface of the carbonyl iron powder and the interaction between the carbonyl iron powder and rubber matrix were investigated. The effects of the silane coupling agent and vulcanization systems on the mechanical properties and damping properties of MRE were evaluated by a dynamic mechanical test, and possible mechanisms are proposed.

## 2. Experimental

### 2.1. Materials

Natural rubber (NR, RSS1#) was used as the matrix in this work, supplied from Indonesia. The carbonyl iron (CI, type SM) particles had an average diameter of 3.4 μm. Accelerators CZ (*N*-cyclohexyl-2-benzothiazole-solenoid) and sulfur (S) were selected as the vulcanization system, and were supplied by the Shanghai Kelong Chemical Co. Ltd. (Shanghai, China). In the CV system, the amounts of CZ and S were 0.5 and 2.5 phr, respectively. In the semi-EV system, the amounts of CZ and S were 1.5 and1.5 phr, respectively. For the EV system, the amounts of CZ and S were 2.5 and 0.5 phr, respectively. Silane coupling agent Si69 was obtained from Nanjing Daoning Chemical Co. Ltd. (Nanjing, China). The chemical structures of Si69 are listed in Scheme 1. Other ingredients such as activator ZnO, stearic acid, anti-oxidant polymerized 2, 2, 4-trimethyl-1, 2-dihydroquinoline (RD), *N*-isopropyl-*N*′-phenyl-p-phenylenediamine (4010NA), and plasticizer cumarone were also purchased from Shanghai Kelong Chemical Co. Ltd. (Shanghai, China). Table 1 shows the detailed compositions of the MRE samples.

### 2.2. Surface Modification of Carbonyl Iron (CI) Particles

The amount of silane coupling agent was 2% by weight of the CI particles. An aqueous alcohol solution containing 95% ethanol was used, and the pH of the solution was adjusted to 4.0–4.5 with hydrochloric acid. First, Si69 was dispersed in an aqueous alcohol solution at a ratio of 1:100, and stirred into a uniform solution. Then, the CI particles were added to the solution and stirred at 80 °C for 4 h to a uniform distribution of the coupling agents on the surface of the CI particles. The mixture was filtered and washed three times with ethanol. Finally, the modified CI particles were dried at 60 °C in an oven until a constant weight was achieved. In this work, the CI particles modified with the Si69 were named Si-CI particles.

### 2.3. Preparation of Magnetorheological Elastomer (MRE) Samples

The fabrication of a MRE is composed of the following two processes: mixing and vulcanization. First, rubber, CI particles, and other additives were mixed by a two-roll mill (LN-6, Guangdong Lina Co. Ltd., Guangdong, China). The details of the mixing process can be obtained from a previous work. Twenty-four hours later, the resulting mixture was vulcanized on a flat vulcanizer (XLB-D, Qingdao Xueqing Rubber Machinery Co. Ltd., Qingdao, China) under the conditions of 143 °C, 12 Mpa, and an external magnetic field of 1000 mT. The external magnetic field was provided by a self-developed electromagnetic device (shown in Figure 1). The curing time was determined by a curing rheometer (MDR-2000E, Wuxi Liyuan Chemical Equipment Co. Ltd., Wuxi, China). After vulcanizing, the MRE samples were completed. Two groups of MRE samples were prepared. Regardless of the vulcanization system, the sample filled with pure CI particles was named MRE, and the sample with Si-Cl particles was named Si-MRE.

### 2.4. Characterizations

#### 2.4.1. Surface Hydrophobicity Behavior

The surface hydrophobicity of the CI particles was investigated using a contact angle meter (OCA20, Dataphysics Instruments GmbH, Filderstadt, Germany). In order to perform measurements, the material in powder form had to be compacted and then the water was dropped on the sample surface to make a measurement of the contact angle.

#### 2.4.2. Scanning Electron Microscopy

The fracture surface of the MRE samples were observed using a SU3500 scanning electron microscope (SEM) (Hitachi High-Technologies Corporation, Hitachinaka, Japen). The sample was cut perpendicular to the circular surface in liquid nitrogen, and then the fracture surface was coated with a thin layer of gold. Finally, the microstructure of the MRE was observed under an accelerating voltage of 15 kV.

#### 2.4.3. Crosslink Density

Determination of the cross-link solidness of the MRE was acquired by the swelling test. Samples with a thickness of 3 mm and 10 mm in diameter were weighed after being immersed under toluene for 72 h at 30 °C in the dark. Afterward, overstock liquid was removed by blotting with filter paper on the surface of the specimens. Eventually, the specimen was dried at 50 °C until the samples achieved constant weight. For every point, three specimens were produced and average values were calculated. Cross-link density can be computed by the Flory–Rehner equation like those illustrated in Equations (1) and (2).
(1)$$V_{e}=-\frac{1}{v}\left[\frac{\ln\left(1-v_{2}\right)+v_{2}+\chi v_{2}^{2}}{v_{2}^{1/3}}\right]$$
(2)$$v_{2}=\frac{m_{dry}/\rho }{m_{dry}/\rho +\left(m_{wet}-m_{dry}\right)/\rho_{s}}$$
where *V_e_*, *v*_2_, and *v* are the crosslinking density of thee vulcanizate sample, volume fraction of the rubbery phase in swollen vulcanizate, and the molar volume of toluene (*v* = 106.4 cm^3^/mol), respectively. *χ* is the interaction parameter of the rubber network-solvent (*χ* of NR = 0.393). *m_wet_* is the swollen equilibrium mass; *m_dry_* is the dry MREs mass; ρ is the density of rubber (*ρ* = 0.92 g/cm^3^); and *ρ_s_* is the density of the swelling solvent (*ρ_s_* = 0.87 g/cm^3^).

#### 2.4.4. Dynamic Mechanical Properties

The dynamic properties of the MRE were measured using a rheometer (MCR 302, Anton Paar, Ostfildern, Germany) with MR equipment (MRD 70 Physica, Anton Paar, Ostfildern, Germany), which produces fields of equal magnetic perpendicular to the surface of the sample. The principle of the rheometer is demonstrated in Figure 2. The oscillation frequency was 10 Hz, the strain amplitude was 0.02%, and the magnetic field scanning range was 0 mT to 800 mT by adjusting the DC power supply. The measurements were performed in the linear viscoelastic regime. Isotherm VT2 was used to control the temperature at 25 °C In the experiment, a cylinder with a diameter of 20 mm and thickness of 1 mm was set between the rotation disk and the parallel base. When the rotating disk is rotated, the MRE sample is deformed in the oscillation shear mode. At the same time, the rheometer can gauge the amplitude, phases of the output force, calculate the shear storage modulus G′, shear loss modulus G″, and loss factor. In this work, the initial normal force was set to 20 N. Each sample was tested five times at the same condition.

## 3. Results and Discussion

### 3.1. Surface Hydrophobicity of the CI Particles

As illustrated in Figure 2, the contact angles of the water droplet on pure CI particles and Si-CI particle disks were measured at room temperature to obtain the evidence for the coupling of CI particles with silane and to clarify the effects of surface modification, then the hydrophobic CI particles and the hydrophobicity of CI particles. Compared to pure CI particles (Figure 2a), the contact angles of the Si-CI particles (Figure 2b) increased from 51.8° to 120.4° in water, thereby suggesting the increase in surface hydrophobicity and the decrease in surface free-energy of the CI particles. The results presented good accordance with the trend when silane treated other fillers such as silica [26], coal powder [27], and micro fibrillated cellulose [28]. Changes in contact angle were likely to be caused by the hydrophobic functional groups of the silane coupling agent, resulting in an increase in surface hydrophobicity of the modified CI particles, which was also beneficial in improving the dispersibility in the rubber matrix [29].

### 3.2. Morphology

As shown in Figure 3, SEM was used to observe the microstructures of the MRE and Si-MRE samples of the CV system and EV system. In the vulcanization system, the unmodified CI particles (white dots, see Figure 3a,c) were pulled out from the rubber matrix and the gap between the CI particles and rubber, suggesting weak-interaction between the CI particles and rubber. After the modification, the interface between the modified CI particles and rubber matrix became blurrier (see Figure 3b,d), providing indirect proof that the interfacial compatibility was improved. Due to the application of a magnetic field during the vulcanization process, the CI particles were assembled into chain-like structures in the direction of the magnetic fields. Some CI particle aggregations were found in thee MRE. The modified CI particle in Si-MRE was more uniformly dispersed in the rubber matrix, and the chain-like structures were long and straight. Apparently, surface modification resulted in a decrease in the surface energy of CI particles, an enhancement of the space steric effect, and a decrease in aggregation levels, which finally improved the dispersion of the CI particles.

SEM demonstrated that the vulcanization system affected the microstructure of the MRE samples and showed that in aligned particle (white dots) chains in samples of EV systems and CV systems, the particle chain in the EV system samples were more aligned and longer. Separately, the vulcanization system was changed from CV to semi-EV and EV, decreasing the cross-link density for the rubber matrix. The hindrance for its movement of the particles was also reduced, so the advanced chain-like structures would be lightly formed during vulcanization [30,31].

Table 2 shows the MRE and Si-MRE samples with different vulcanization systems. The addition of silane coupling agents reduced the minimum torque ML of all samples due to the plasticizing effect. The values of the maximum torque MH and torque difference ΔM of all system samples showed an increase after adjunction [32]. During the vulcanization process, the tetranuclear group of Si69 generates sulfur radicals after dissociating. Sulfur free radicals cross-link with the rubber molecules and exhibited an increase in torque macroscopically under the action of the accelerator. Si69 could also bring an increase in torque. Si69 contained ethoxy hydrolyzable groups at both ends, bonding with CI particles and the tetra sulfide group in the center, which could increase the interaction between the particles and matrix.

The addition of silane coupling agents also affected the scorch time and vulcanization time of samples of different systems. The scorch time of samples of the CV system decreased, and the vulcanization time increased. Both the scorch time and the vulcanization time of the semi-EV system samples increased. The scorch time and vulcanization time of thee EV system were reduced. The results proved that the sulfur radicals decomposed by Si69 participated in the vulcanization reaction.

### 3.3. Dynamic Mechanical Analysis

The mechanical properties and the MR effect of MRE were evaluated by investigating the shear storage modulus (*G′*) by defining *G′*_0_ as the zero-field modulus, and *G_s_* as the saturated magnetic field modulus. Thus, the ∆*G′_s_* corresponded to define the saturated magnetic-induced coefficient as the value of *G′_s_* minus *G′*_0_ and the MR effect was defined as ∆*G′_s_*/*G′*_0_.

The G′_0_ of the zero-field modulus of MRE and Si-MRE with various vulcanization systems were performed in Figure 4. This indicated that in all the vulcanization systems used, the *G′*_0_ of Si-MRE was higher than that of the MRE. MREs are particle-reinforced composites whose strength depend primarily on the success of the applied load transmitted to the filler through the matrix [7,33]. The introduction of the silane coupling agents promoted the adhesion of the matrix to the filler, thus improving the mechanical properties of the MRE samples.

Figure 4 also showed the dependence of the zero-field modulus *G′*_0_ on the vulcanization system. In the presence of the silane coupling agent, the *G′*_0_ of CV systems, semi-EV systems, and EV systems showed a tendency to increase sequentially. For MRE, matrix modulus, particle distribution, and matrix properties also played a key role in the mechanical properties. It has been shown that the higher-order of magnetic particles in the rubber matrix have a stronger blocking effect, resulting in higher mechanical properties of the MRE [33,34]. By comparing Figure 3a,b with Figure 3c,d, it was found that the particle chains in the EV system samples were longer and more aligned, while the particle chains in the CV system samples were thicker and shorter. This explained why the zero-field modulus *G′*_0_ tended to decrease in the following sequence: EV, semi-EV, and CV. However, it was worth noting that the effect of silane on the *G′*_0_ was highly dependent on the vulcanization system. The effect of silane coupling agents was more obvious in the EV system compared with the semi-EV and CV systems. Si69 possesses four sulfur atoms in its molecule (see Scheme 1), and these sulfurs could participate in the crosslink of rubber molecules in the presence of an accelerator during the vulcanization reaction [32]. As a result, the phenomenon was apparent only in the EV system that had a large amount of accelerator and little free sulfur. Therefore, the *G′*_0_ of the EV system was increased more than that of the semi-EV and EV systems. In addition, the phenomenon above-mentioned in this work was defined as the “sulfur contribution effect”.

When a certain magnetic field strength was applied to the MRE sample, the CI particles were easily magnetized and aligned in the direction of the magnetic field due to the induced magnetic dipole–dipole interaction. This was the main reason for the change in shear modulus [35]. Figure 5 shows the saturated magnetic-induced modulus ∆*G′_s_* of MRE and Si-MRE with various vulcanization systems. As expected, the saturated magnetic-induced modulus ∆*G′_s_* of Si-MRE was higher than that of the MRE, which might be attributed to the better rubber–particle interaction and improved particle dispersion [13]. Evidence of the above reason was obtained from the SEM micrographs shown in Figure 3. The modified CI particles were more strongly bonded to the matrix, and the particle chains were more aligned in the direction of the magnetic field, which resulted in an increase in the ∆*G′_s_* of the MRE sample. The results might also be related to the sulfur contribution effect of silane. After the introduction of additional sulfur, the stiffness of the matrix of the EV system was greatly increased, so that the CI particles were easier to introduce the stress into the rubber matrix under the action of the magnetic fields, and thus the ∆*G′_s_* of the MRE sample was also higher.

To further explain the effect of sulfur contribution, the crosslink density of the MRE and Si-MRE of different vulcanization systems was investigated, as shown in Figure 6. It could be observed that the silane coupling agent had no significant effect on the crosslink density of the CV and semi-EV system samples. However, in the EV system samples, the effect of silane on the crosslink density was more pronounced. The result clearly confirmed that the effect of the sulfur contribution of Si69 played an important role in the EV system.

The results also revealed the dependence of the magnetic-induced modulus on the vulcanization system. In the presence of silane, the EV system demonstrated the highest ∆*G′s*, followed by the semi-EV and CV systems separately. The chain-like structures were more simple to form by the rubber matrix of the EV system. In this situation, the extra interaction with particles would be introduced to the matrix. Accordingly, the EV system represented the highest ∆*G′s*.

Figure 7 shows the Payne effect factor of the pure-MRE and Si-MRE of different vulcanization systems under a magnetic field of 800 mT. The Payne effect was weakened and the Payne effect factor of MRE increased when compared with the increase at the zero field, while the Payne effect factor of Si-MRE showed a decrease. For example, the Payne effect factor of the MRE of the CV system was increased to 0.497, and that of the Si-MRE was reduced to 0.456. The storage modulus of the Si-MRE in the linear viscoelastic region (M0) increased more than the modulus of MRE after applying a magnetic field. The storage modulus under (Ms) was almost the same, and the Payne effect factor of Si-MRE was relatively reduced.

Effects of the silane coupling agents on the MR effect of the MRE and Si-MRE with various vulcanization systems are displayed at Figure 8. The effect of MR of the Si-MRE was unexpectedly lower than that of the MRE. The reason was that the introduction of silane not only improved the interaction between CI particles and matrix and the degree of orientation of the CI particles, but also increased the zero-field modulus of the MRE sample. Thereby, the MR effect of the MRE filled with silane-modified CI particles was lower than that of the MRE filled with the unmodified CI particles. This was disadvantageous to the application of the MRE.

Taking into consideration the effects on the vulcanization system, the highest value of the MR effect was acquired in the CV system, followed by the EV and semi-EV systems, separately. The chain-like structures simultaneously enhanced the zero-field modulus *G′*_0_ and the saturant magnetic-induced shear modulus ∆*G* of the compound, causing the lower MR effect in the EV systems than the CV systems.

Figure 9 shows the loss factors of the MRE and Si-MRE with different vulcanization systems under different magnetic fields. The loss factor determined the energy dissipation of materials. It could be seen that the loss factor of all samples decreased with the increasing magnetic fields, which was in agreement with the results reported by other researchers [11,36]. It could also be seen that the loss factor of the Si-MRE was lower than that of the MRE. Due to the functionalization of the surface of the CI particles, there was a better combination between the modified CI particles and the matrix. Therefore, the relative slip of the chain molecules were reduced, resulting in lower energy dissipation [2]. Furthermore, the increase in the dispersion of CI particles in the rubber matrix (see SEM micrographs of Figure 3) can also reduce the loss factor of the MRE [37]. In addition, the effect of silane still depended on the vulcanization system. After the introduction of silane, the changes in the loss factor were more pronounced in the EV system. Possible explanations were still related to the sulfur contribution effects.

### 3.4. Enhancement Mechanism of Silane Coupling Agent Si69

The mechanism of the interaction between the rubber matrix and CI particles is shown in Figure 10. Under shear stress, the rubber matrix moved in the direction of the shear force while the CI particles tended to maintain a chain structure in the vertical direction. The rubber matrix and CI particles began to slip. As could be seen in Figure 10a, the unmodified CI particles agglomerated due to the higher surface free energy. The interaction between the CI particles and rubber matrix was weak, and the rubber matrix was easily bonded from the particle surface. When shear stress was applied, the sliding friction between the CI particles and rubber matrix was huge, so the energy dissipation tended to be bigger and the loss factor was larger. After the CI particles were modified (see Figure 10b), the interaction between the CI particles and rubber matrix was enhanced, and the aggregation was also lowered, which improved the zero-field modulus and saturated magnetic-induced modulus of the MRE samples.

The reaction scheme for forming a chemical bond between the CI particles and rubber matrix through silane is shown in Figure 11. First, in an aqueous ethanol solution under acid-catalyzed conditions, the ethoxy groups of Si69 were hydrolyzed to form silanol groups (–Si–O–OH). Subsequently, the silanol groups were dehydrated with hydroxyl groups on the surface of the CI particle to form a covalent bond, resulting in an increase in the hydrophobicity of CI particles and particle aggregation of CI particles. Finally, the sulfide linkage of Si69 bonded to the CI particles was dissociated to form sulfur radicals. The sulfur radicals could participate in the vulcanization reaction of rubber, resulting in a change in the crosslink structures of the rubber matrix. The degree of the change was more apparent in the EV system with lower sulfur content.

## 4. Conclusions

Effects of the silane coupling agents on the properties of MREs with various vulcanization systems were investigated. The water contact angle test suggested that the hydrophobicity of the surface of CI particles was significantly improved by silane. The SEM analysis showed that the interfacial interaction between modified CI particles and the rubber matrix with various vulcanization systems was also improved. The results of the dynamic mechanical properties test showed that after modifying, the zero-field modulus and saturated magnetic-induced modulus of the MRE increased and the loss factor decreased. In particular, these changes of dynamic mechanical properties were more obvious in the EV system. However, the addition of silane also resulted in a decrease in the MR effect of all systems, which was detrimental to the application of MRE. The enhancement mechanism of silane was studied and it was found that silane could increase the interaction between the carbonyl iron powder and the rubber matrix since it formed a chemical bond between them. The chemical bond associated with the matrix was formed by the sulfur radical, which was generated by the hydrolysis of the silane participating in the sulfurization reaction and affected the crosslink structures of the matrix. The effects of the silane were more obvious in the EV system with less sulfur content.

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
