# Peer review of "Reinforcing Behaviors of Sulfur-Containing Silane Coupling Agent in Natural Rubber-Based Magnetorheological Elastomers with Various Vulcanization Systems"

_materials, 2020, doi:10.3390/ma13225163_

Round 1

Reviewer 1 Report

Nice attempt to improve the interfacial interactions between CIP particles and NR matrix through silanization for the production of reinforced MREs. Please address the following issues before this work can be considered for publication:

The surface chemistry of unmodified and silane-treated CIP particles must be chemically analysed by using (may be) XPS method, which can be useful in supporting the observed improvements in material characteristics.

Please show the cure curves of MREs for understanding the effect of different sulphur vulcanization systems. The effect on the processing safety, developed torque and cure reversion may be notable for the anisotropic MREs.

The interpretation of SEM images appears too speculative; the discussion about interfacial compatibility is not convincing.

Figure 5 and 6 are same with two different captions.

The calculation of crosslink density by Flory-Rehner equation was mentioned under experimental section but the results are not shown in the whole manuscript?

The illustration of Figure 10 is not supported by any chemical analysis and structural characterization method.

Author Response

Thanks for your valuable report. Please check the responses in the attached file.

Reviewer 2 Report

The work done by M. Wang et al pertains to the development of magnetorheological elastomers based on natural rubber and carbonyl iron magnetic microparticles. The focus is on the effects of particle surface modification and vulcanization process on the viscoelastic properties of the resulting composites. Sulfur-containing silane coupling agent is chosen as a particle modifier. Various vulcanization processes are examined differing in the amount of reaction accelerator and sulfur. It has been shown that MREs based on modified iron particles demonstrated larger initial zero-field values of the storage modulus but a lower relative magnetorheological effect. The influence of the silane modification on the storage and loss moduli of MREs were more pronounced for the vulcanization process with less sulfur content. The obtained results have a certain value for technological issues concerning synthesis of MREs. However, a considerable revision of the manuscript is necessary before its publication. I would recommend the authors to consider the following points:

  1. First of all, the text should be significantly improved, not only bad English, which makes it difficult to understand some sentences, but also many typos.
  2. Figs. 5 and 6 contain similar plots! Fig. 6 should show the crosslink density instead of storage modulus!
  3. Since the focus is on particle surface modification effects, a thorough characterization of the particles before and after modification should be performed. Contact angle tests are not enough, they do not directly show the degree of surface modification, microscopic studies (scanning electron microscopy, for instance) could be helpful to estimate the degree of surface modification.
  4. The protocol of the rheological measurement should be given in detail. What was the osciliation frequency and strain amplitude? Whether the measurements have been performed in the linear viscoelastic regime? Whether the surface modification and type of vulcanization process affect the Payne effect which is typical for MRE? The latter is very important for MRE damping applications.
  5. In section Preparation of MREs it is stated that “The curing time was determined from the curing rheometer” which is not an explanation at all.
  6. What are the measurement errors of G’ and G’’? they should be indicated in Figs.4-8.
  7. The differences in SEM images shown in Fig.3 are not obvious, they look quite similar. Any conclusions that “the particle chains in the EV system sample were longer and more aligned” are not obvious.

Author Response

(The authors gave the same response as above.)

Round 2

Reviewer 1 Report

Most of the recommendations seem to have been incorporated in the revised version of manuscript, which can be accepted for publication.

Author Response

Thanks for your valuable comments.

Reviewer 2 Report

The manuscript has been improved but not all reviewer recommendations have been taken into account. In particular, I could not find any details on the rheological measurements in the revised manuscript. Furthermore, the authors have added a graph demonstrating "Payne effect factors" but they do not explain how these factors are defined. English should be considerably improved!
